# POLICY-DRIVEN WORLD MODEL ADAPTATION FOR ROBUST OFFLINE MODEL-BASED REINFORCEMENT LEARNING

## ABSTRACT

Offline reinforcement learning (RL) offers a powerful paradigm for data-driven control. Compared to model-free approaches, offline model-based RL (MBRL) explicitly learns a world model from a static dataset and uses it as a surrogate simulator, improving data efficiency and enabling potential generalization beyond the dataset support. However, most existing offline MBRL methods follow a two-stage training procedure: first learning a world model by maximizing the likelihood of the observed transitions, then optimizing a policy to maximize its expected return under the learned model. This objective mismatch results in a world model that is not necessarily optimized for effective policy learning. Moreover, we observe that policies learned via offline MBRL often lack robustness during deployment, and small adversarial noise in the environment can lead to significant performance degradation. To address these, we propose a framework that dynamically adapts the world model alongside the policy under a unified learning objective aimed at improving robustness. At the core of our method is a maximin optimization problem, which we solve by innovatively utilizing Stackelberg learning dynamics. We provide theoretical analysis to support our design and introduce computationally efficient implementations. We benchmark our algorithm on twelve noisy D4RL MuJoCo tasks and three stochastic Tokamak Control tasks, demonstrating its state-of-the-art performance.

## 1 INTRODUCTION AND RELATED WORKS

Offline RL (Levine et al., 2020) leverages offline datasets of transitions, collected by a behavior policy, to train a policy. To avoid overestimation of the expected return for out-of-distribution states, which can mislead policy learning, model-free offline RL methods (Kumar et al., 2020; An et al., 2021) often constrain the learned policy to remain close to the behavior policy. However, acquiring a large volume of demonstrations from a high-quality behavior policy, can be expensive. This challenge has led to the development of offline model-based reinforcement learning (MBRL) approaches, such as Yu et al. (2020); Sun et al. (2023); Chen et al. (2024c). These methods train dynamics models from offline data and optimize policies using imaginary rollouts simulated by the models. Notably, the dynamics modeling is independent of the behavior policy, making it possible to learn effective policies from any behavior policy that reasonably covers the state-action spaces.

As detailed in Section 2, offline MBRL typically follows a two-stage framework: first, learning a world model by maximizing the likelihood of transitions in the offline dataset; and second, using the learned model as a surrogate simulator to train RL policies. Unlike in online MBRL (Ross & Bagnell, 2012; Hafner et al., 2020; Chen et al., 2023a), the world model in offline settings is typically not adapted alongside the policy. Moreover, the model training objective, i.e., likelihood maximization, differs from the goal of policy optimization, i.e., maximizing the expected return, leading to the issue of objective mismatch (Wei et al., 2024). There has been extensive research addressing this mismatch in online MBRL (e.g., Farahmand (2018); Grimm et al. (2020); Nikishin et al. (2022); Eysenbach et al. (2022); Ma et al. (2023); Vemula et al. (2023)), primarily by directly optimizing the model to increase the current policy's return in the environment. However, applying these strategies in the offline setting can be problematic. Since the real environment is inaccessible during offline training, optimizing the world model solely to increase returns in imagined rollouts can cause it

to diverge from the true dynamics, thereby misleading the policy updates. Comparisons between Yang et al. (2022) and Sun et al. (2023) show that, for offline MBRL, return-driven model adaptation can underperform approaches that do not dynamically update the world model. **How to adapt the world model alongside the policy under a unified objective remains an open challenge in offline MBRL.** In this paper, we propose such a unified training framework to learn robust RL policies from offline data. Specifically, the objective is formulated as a maximin problem: we maximize the worst-case performance of the policy, where the policy seeks to maximize its expected return while the world model is updated adversarially to minimize it. This direction of model updating is opposite to that used in online MBRL, as conservatism is critical in offline RL (Levine et al., 2020).

Another motivation behind our algorithm design is that the state-of-the-art model-free and model-base offline RL algorithms lack robustness during deployment, as demonstrated in Figure 1, as the learned policies often overfit to the static dataset or data-driven dynamics models. Changing the objective to maximizing the worst-case performance of the policy can potentially mitigate such issues. While several theoretical studies have investigated robustness in (model-based) RL (Huang et al., 2022; Panaganti et al., 2022; Uehara & Sun, 2023; Shi & Chi, 2024), the proposed algorithms are not scalable to high-dimensional continuous control tasks. The work most closely related to ours is RAMBO (Rigter et al., 2022), which also targets robust offline MBRL. However, RAMBO does not formulate the problem as a Stackelberg game, where the policy and world model act as leader and follower in a general-sum game. Also, it does not

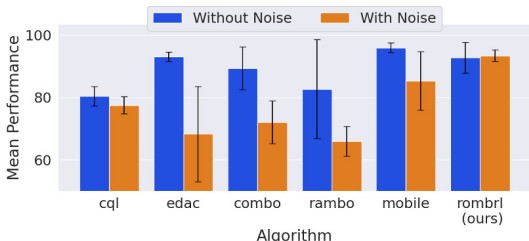

Figure 1: Average scores of different offline RL algorithms on nine D4RL MuJoCo tasks (corresponding to the tasks in Table 1 excluding those with the random data type), before and after applying random noise to state transitions. The noise is modeled as zero-mean Gaussian with a standard deviation equal to 5% of the state change, simulating common measurement noise.

employ Stackelberg learning dynamics (Rajeswaran et al., 2020; Fiez et al., 2020; Zhou & Qu, 2023; Chen et al., 2023c), which are essential for solving the maximin optimization inherent in robust RL (see Section 2 for details). Additionally, since the world model is updated jointly with the policy, historical rollouts in the replay buffer would become outdated for policy training due to distributional shift. While RAMBO overlooks this issue, we introduce a novel gradient mask mechanism (detailed in Appendix J) to mitigate it while preserving training efficiency.

Our contributions are as follows: (1) We propose a novel offline MBRL algorithm (ROMBRL) that jointly optimizes the model and policy for robustness by solving a constrained maximin optimization problem. We further provide novel theoretical guarantees on the suboptimality gap of the resulting robust policy. (2) To solve this maximin problem, we model it as a Stackelberg game and introduce novel Stackelberg learning dynamics for stochastic gradient updates of the policy and model. To the best of our knowledge, this is the first application of the Stackelberg game framework to offline MBRL. (3) We develop practical implementations of the proposed learning dynamics, including the use of the Woodbury matrix identity for efficient second-order gradient computation and a gradient-mask mechanism that enables off-policy training to improve data efficiency. (4) ROMBRL achieves state-of-the-art performance and robustness on the widely-used D4RL MuJoCo benchmark under noisy conditions and on a challenging Tokamak Control benchmark for nuclear fusion. In ideal noiseless settings, its performance is on par with SOTA algorithms and significantly surpasses RAMBO, another baseline designed for robustness.

## 2 BACKGROUND

**Offline Model-based Reinforcement Learning:** A Markov Decision Process (MDP) (Puterman, 2014) can be described as a tuple $M = \langle \mathcal{S}, \mathcal{A}, P, R, d_0, \gamma \rangle$. $\mathcal{S}$ and $\mathcal{A}$ are the state space and action space, respectively. $P : \mathcal{S} \times \mathcal{A} \to \Delta_{\mathcal{S}}$ represents the transition dynamics function, $R : \mathcal{S} \times \mathcal{A} \to \Delta_{[0,1]}$ defines the reward function, and $d_0 : \mathcal{S} \to \Delta_{\mathcal{S}}$ specifies the initial state distribution, where $\Delta_{\mathcal{X}}$ is the set of probability distributions over the space $\mathcal{X}$. $\gamma \in [0, 1)$ is a discount factor.

Given an offline dataset of transitions $\mathcal{D}_\mu = \{(s_i, a_i, r_i, s_i')_{i=1}^N\}$ collected by a behavior policy $\mu$, typical offline MBRL methods first learn a world model $P_\phi, R_\phi$ through supervised learning:

$$\max_\phi \mathbb{E}_{(s,a,r,s')\sim\mathcal{D}_\mu} \left[\log P_\phi(s'|s,a) + \log R_\phi(r|s,a)\right] \tag{1}$$

Then, a policy $\pi_\theta : \mathcal{S} \to \Delta_{\mathcal{A}}$ is trained to maximize the expected return $J(\theta, \phi)$ in the MDP $M_\phi = \langle \mathcal{S}, \mathcal{A}, P_\phi, R_\phi, d_0, \gamma \rangle$: ($V(s'; \theta, \phi) = \mathbb{E}_{a'\sim\pi_\theta(\cdot|s')} [Q(s', a'|\theta, \phi)]$ is the value function.)

$$\max_\theta \mathbb{E}_{s\sim d_0(\cdot), a\sim\pi_\theta(\cdot|s)} [Q(s, a; \theta, \phi)], \; Q(s, a; \theta, \phi) = \mathbb{E}_{r\sim R_\phi(\cdot|s,a), s'\sim P_\phi(\cdot|s,a)} \left[r + \gamma V(s'; \theta, \phi)\right] \tag{2}$$

To address uncertainty of the learned world model, offline MBRL methods (Chen et al., 2024a) usually learn an ensemble of world models from $\mathcal{D}_\mu$ and apply an ensemble-based reward penalty to $r$ in Eq. (2), discouraging the agent from exploring regions where predictions from the ensemble members exhibit high variance. However, these reward penalty terms are heuristic-based, and as a result, these algorithms lack formal performance guarantees (Yu et al., 2021).

**Learning Dynamics in Stackelberg Games:** In a Stackelberg game, the leader (who plays first) and follower aim to solve the following two optimization problems, respectively:

$$\min_{x_1\in\mathcal{X}_1} \{f_1(x_1, x_2^*) \mid x_2^* \in \arg\min_{x_2\in\mathcal{X}_2} f_2(x_1, x_2)\}, \; \min_{x_2\in\mathcal{X}_2} f_2(x_1, x_2) \tag{3}$$

The authors of Fiez et al. (2020) propose update rules for the leader and follower in a class of two-player smooth games defined on continuous and unconstrained $\mathcal{X}_1, \mathcal{X}_2$. Specifically, at each iteration $k$, $x_1$ and $x_2$ are updated as follows:

$$\begin{aligned}x_1^{k+1} &= x_1^k - \eta_1^k \omega_1^k, \; \omega_1^k = D_1 f_1(x^k) - D_{21} f_2(x^k)^T (D_{22} f_2(x^k))^{-1} D_2 f_1(x^k); \\ x_2^{k+1} &= x_2^k - \eta_2^k \omega_2^k, \; \omega_2^k = D_2 f_2(x^k).\end{aligned} \tag{4}$$

Here, $x^k$ denotes the value of $(x_1, x_2)$ at iteration $k$, while $(\eta_1^k, \eta_2^k)$ represent the learning rates; $D_1 f_1(x^k)$ represents the partial derivative of $f_1(x^k)$ with respect to $x_1$, and similarly for other derivatives. The expression for $\omega_1^k$ is derived based on the total derivative of $f_1(x_1, x_2^*)$ with respect to $x_1$, of which the details are available in Appendix A.

The learning target for Stackelberg games is to reach Local Stackelberg Equilibrium (LSE (Başar & Olsder, 1998)):

**Definition 1** (LSE). *Consider $U_i \subset \mathcal{X}_i$ for each $i \in \{1, 2\}$. The strategy $x_1^* \in U_1$ is a local Stackelberg solution for the leader, if $\forall x_1 \in U_1$, $\sup_{x_2\in R_{U_2}(x_1^*)} f_1(x_1^*, x_2) \leq \sup_{x_2\in R_{U_2}(x_1)} f_1(x_1, x_2)$, where $R_{U_2}(x_1) = \{y \in U_2 \mid f_2(x_1, y) \leq f_2(x_1, x_2), \forall x_2 \in U_2\}$. Moreover, $(x_1^*, x_2^*)$ for any $x_2^* \in R_{U_2}(x_1^*)$ is a Local Stackelberg Equilibrium on $U_1 \times U_2$.*

When using unbiased estimators $(\hat{\omega}_1^k, \hat{\omega}_2^k)$ in place of $(\omega_1^k, \omega_2^k)$ in the iterative updates, Theorem 7 in Fiez et al. (2020) establishes that there exists a neighborhood $U = U_1 \times U_2$ of the LSE $x^* = (x_1^*, x_2^*)$ such that for any $x^0 \in U$, $x^k$ converges almost surely to $x^*$. This result holds for a smooth general-sum game $(f_1, f_2)$, where player 1 is the leader and $\eta_1^k = o(\eta_2^k)$, under the standard stochastic approximation conditions: $\sum_k \eta_i^k = \infty, \sum_k (\eta_i^k)^2 < \infty, \forall i$.

## 3 THEORETICAL RESULTS

Our method is based on an robust offline MBRL objective[1]: (For simplicity, we merge $P_\phi$ and $R_\phi$ into a single notation, $P_\phi(r, s'|s, a)$, in the following discussion.)

$$\max_\theta J(\theta, \phi'), \; s.t., \; \phi' \in \arg\min_{\phi\in\Phi} J(\theta, \phi) \tag{5}$$

The uncertainty set $\Phi = \{\phi \in \mathcal{M} \mid \mathbb{E}_{(s,a)\sim\mathcal{D}_\mu, (r,s')\sim P_{\bar{\phi}}(\cdot|s,a)} \left[\log P_{\bar{\phi}}(r, s'|s, a) - \log P_\phi(r, s'|s, a)\right] \leq \epsilon\}$, where $\bar{\phi}$ is an optimal solution to Eq. (1) (i.e., a maximum likelihood estimator). **Intuitively, $\pi_\theta$ is trained to maximize its worst-case performance within an uncertainty set of world models, ensuring robust deployment performance.**

---

[1]CPPO-LR, proposed in Uehara & Sun (2023), adopts a similar objective function; however, its theoretical analysis – specifically Lemma 6 and Lemma 7 in its Appendix B.2 – is incorrect.

Define $d_{\theta^*,\phi^*}(s,a) = (1-\gamma)\sum_{t=0}^{\infty}\gamma^t d_{\theta^*,\phi^*}^t(s,a)$, where $\theta^*$ and $\phi^*$ represent the parameters of a comparator policy and the true MDP, respectively, and $d_{\theta^*,\phi^*}^t(s,a)$ denotes the probability density of the agent reaching $(s,a)$ at time step $t$ when following $\pi_{\theta^*}$ in the environment $M_{\phi^*}$. Further, as in Uehara & Sun (2023), we can define a concentrability coefficient $C_{\phi^*,\theta^*} = \sup_{\phi \in \mathcal{M}} \frac{\mathbb{E}_{(s,a)\sim d_{\theta^*,\phi^*}(\cdot)}\left[TV(P_\phi(\cdot|s,a),P_{\phi^*}(\cdot|s,a))^2\right]}{\mathbb{E}_{(s,a)\sim d_{\mu,\phi^*}(\cdot)}\left[TV(P_\phi(\cdot|s,a),P_{\phi^*}(\cdot|s,a))^2\right]}$, where $TV(\cdot,\cdot)$ denotes the total variation distance between two distributions. Then, we have the following theorem[2]: (Please refer to Appendix B for the proof.)

**Theorem 1.** *Assume $\phi^* \in \Phi$ with probability at least $1 - \delta/2$. Then, for any comparator policy $\pi_{\theta^*}$, with probability at least $1 - \delta$, the performance gap in expected return between $\pi_{\theta^*}$ and $\pi_{\hat{\theta}}$ satisfies:*

$$J(\theta^*,\phi^*) - J(\hat{\theta},\phi^*) \leq \frac{\sqrt{C_{\phi^*,\theta^*}}}{(1-\gamma)^2}\sqrt{4\epsilon + c\left(\sqrt{\frac{\log(2|\Phi|/\delta)}{N}} + \frac{\log(2|\Phi|/\delta)}{N}\right)}, \qquad (6)$$

*where $N$ and $|\Phi|$ denote the size of $\mathcal{D}_\mu$ and $\Phi$, respectively, $\hat{\theta}$ is an optimal solution to Eq. (5), and $c$ is a constant.*

Notably, if $\pi_{\theta^*}$ is the optimal policy on $M_{\phi^*}$ and $C_{\phi^*,\theta^*} < \infty$, Eq. (6) represents the suboptimality gap of the learned policy $\pi_{\hat{\theta}}$ through Eq. (5). Moreover, Theorem 1 applies to a general function class of $P_\phi$ and relies on the assumption that $\phi^* \in \Phi$ with probability at least $1 - \delta/2$. By specifying the function class of $P_\phi$, we can determine the uncertainty range $\epsilon$ to ensure that $\Phi$ includes $\phi^*$ with high probability, allowing us to remove this assumption.

Denote $\mathcal{D}_{sa}^\mu$ as the number of unique state-action pairs in $\mathcal{D}_\mu$ and $N_{sa}$ as the number of transitions in $\mathcal{D}_\mu$ that are sampled at $(s,a)$. Further, we define $\widetilde{N} = \max\{N_{sa} \mid (s,a) \in \mathcal{D}_\mu\}$. For tabular MDPs, the world models (i.e., $P_\phi$) follow categorical distributions and we have the following theorem[3]: (Please refer to Appendix C for the proof and Appendix E for a discussion on $|\Phi|$.)

**Theorem 2.** *In tabular MDPs, suppose $K$ is the alphabet size of the world model and $3 \leq K \leq \frac{N_{sa}C_0}{e} + 2$, $\forall(s,a) \in \mathcal{D}_\mu$. Then, $\phi^* \in \Phi$ with probability at least $1 - \delta/2$ when $\epsilon = \frac{\mathcal{D}_{sa}^\mu}{N}\log\frac{2C_1K(C_0\widetilde{N}/K)^{0.5K}\mathcal{D}_{sa}^\mu}{\delta}$.*

Further, most recent offline MBRL methods (Yu et al., 2020; Lu et al., 2022; Sun et al., 2023) utilize deep neural network-based world models and represent $P_\phi(s',r \mid s,a)$ as a multivariate Gaussian distribution with a diagonal covariance matrix, $\forall(s,a)$. In this case, we have the following theorem: (Please check Appendix D for a non-asymptotic representation of $\epsilon$ and the proof.)

**Theorem 3.** *For MDPs with continuous state and action spaces, where the world model follows a diagonal Gaussian distribution, let $d$ denote the dimension of the state space. Then, $\phi^* \in \Phi$ with probability at least $1 - \delta/2$, when $\epsilon = \mathcal{O}\left(\frac{\mathcal{D}_{sa}^\mu d^2}{N}\log\frac{\mathcal{D}_{sa}^\mu d}{\delta}\right)$ as $N_{sa} \to \infty, \forall(s,a) \in \mathcal{D}_\mu$.*

## 4 POLICY-GUIDED WORLD MODEL ADAPTATION FOR ENHANCED ROBUSTNESS

To obtain the theoretical guarantee shown in Section 3, the maximin problem in Eq. (5) needs to be well solved. A straightforward approach, as employed in Rigter et al. (2022), is to update the policy $\pi_\theta$ and world model $P_\phi$ alternatively, treating the other as part of the environment during each update step. Specifically, the training process at iteration $k$ is as follows:

$$\theta^{k+1} = \theta^k + \eta_\theta^k \nabla_\theta J(\theta^k,\phi^k); \ \phi^{k+1} = \phi^k - \eta_\phi^k \nabla_\phi \mathcal{L}(\theta^k,\phi^k). \qquad (7)$$

---

[2]For a discounted finite-horizon MDP with horizon $h$, Theorem 1 remains valid with the following modifications: (1) replacing $\frac{1}{(1-\gamma)^2}$ in Eq. (6) with $\frac{(1-\gamma^h)^2}{(1-\gamma)^2}$ and (2) replacing $d_{\theta^*,\phi^*}(s,a)$ with $d_{\theta^*,\phi^*}^h(s,a) = \frac{1-\gamma}{1-\gamma^h}\sum_{t=0}^{h}\gamma^t d_{\theta^*,\phi^*}^t(s,a)$. The derivation is similar with the one presented in Appendix B.

[3]As defined in Mardia et al. (2020), $C_0 \approx 3.20$ and $C_1 \approx 2.93$. The expression for $\epsilon$ when $K$ falls into different ranges (e.g., $K \geq N_{sa}C_0 + 2$) can be derived similarly based on Theorem 3 from Mardia et al. (2020).

Here, $\mathcal{L}(\theta^k, \phi^k) = J(\theta^k, \phi^k) + \lambda \mathbb{E}_{(s,a) \sim \mathcal{D}_\mu} \left[ KL(P_{\bar{\phi}}(\cdot|s,a) || P_{\phi^k}(\cdot|s,a)) - \epsilon \right]$, $\eta_\theta^k, \eta_\phi^k$ are learning rates at iteration $k$. In this case, the constrained optimization problem $\min_{\phi \in \Phi} J(\theta, \phi)$ is relaxed into an unconstrained optimization problem $\min_{\phi \in \mathcal{M}} \mathcal{L}(\theta, \phi)$, by employing the Lagrangian formulation and treating the Lagrange multiplier $\lambda > 0$ as a hyperparameter. This method is simple but lacks formal convergence guarantees. The alternating updates may lead to instability due to the non-stationarity introduced by treating the other model as part of the environment in separate updates.

By viewing the policy and world model as the leader and follower in a general-sum Stackelberg game, we can apply the Stackelberg learning dynamics (i.e., Eq. (4)) to iteratively update the policy and world model. Such learning dynamics have proven effective in robust online RL (Huang et al., 2022) and offline model-free RL (Zhou & Qu, 2023). In particular, the update at iteration $k$ is as follows:

$$\theta^{k+1} = \theta^k + \eta_\theta^k \left[ \nabla_\theta J(\theta^k, \phi^k) - \nabla_{\phi\theta}^2 \mathcal{L}(\theta^k, \phi^k)^T (\nabla_\phi^2 \mathcal{L}(\theta^k, \phi^k))^{-1} \nabla_\phi J(\theta^k, \phi^k) \right];$$
$$\phi^{k+1} = \phi^k - \eta_\phi^k \nabla_\phi \mathcal{L}(\theta^k, \phi^k). \tag{8}$$

The formula above can be derived by substituting $(x_1, x_2)$ and $(f_1, f_2)$ in Eq. (4) with $(\theta, \phi)$ and $(-J, \mathcal{L})$, respectively. Compared to Eq. (7), the policy updates incorporate gradient information from the world model, as the best-response world model is inherently a function of the policy. This dependency allows the policy optimization process to account for the influence of the evolving world model, leading to potentially more stable learning dynamics. As noted in Section 2, models updated using such learning dynamics are guaranteed to converge to a local Stackelberg equilibrium under mild conditions.

The second approach still solves an unconstrained problem $\min_{\phi \in \mathcal{M}} \mathcal{L}(\theta, \phi)$. As an improvement, we propose a novel learning dynamics, where the follower $\theta$ solves the constrained problem $\min_{\phi \in \Phi} J(\theta, \phi)$ as a response to the leader:

$$\theta^{k+1} = \theta^k + \eta_\theta^k \left[ \nabla_\theta J(\theta^k, \phi^k) - \nabla_{\phi\theta}^2 \mathcal{L}(\theta^k, \phi^k, \lambda^k)^T H(\theta^k, \phi^k, \lambda^k) \nabla_\phi J(\theta^k, \phi^k) \right];$$
$$\phi^{k+1} = \phi^k - \eta_\phi^k \nabla_\phi \mathcal{L}(\theta^k, \phi^k, \lambda^k), \quad \lambda^{k+1} = [\lambda^k + \eta_\lambda^k \nabla_\lambda \mathcal{L}(\theta^k, \phi^k, \lambda^k)]^+. \tag{9}$$

In the equation above, $H(\theta^k, \phi^k, \lambda^k) = A^{-1} + \lambda^k A^{-1} B S^{-1} B^T A^{-1}$, $S = C - \lambda^k B^T A^{-1} B$, $A = \nabla_\phi^2 \mathcal{L}(\theta^k, \phi^k, \lambda^k)$, $B = \nabla_{\phi\lambda}^2 \mathcal{L}(\theta^k, \phi^k, \lambda^k)$, $C = \nabla_\lambda \mathcal{L}(\theta^k, \phi^k, \lambda^k)$, and $\mathcal{L}(\theta^k, \phi^k, \lambda^k) = J(\theta^k, \phi^k) + \lambda^k \mathbb{E}_{(s,a) \sim \mathcal{D}_\mu} \left[ KL(P_{\bar{\phi}}(\cdot|s,a) || P_{\phi^k}(\cdot|s,a)) - \epsilon \right]$ is the Lagrangian. Eq. (9) presents how to update the dual variable in the constrained optimization problem $\min_{\phi \in \Phi} J(\theta, \phi)$, i.e., $\lambda$, alongside $\theta$ and $\phi$. Compared to Eq. (8), the policy updates additionally incorporate gradient information from the best-response dual variable, which is a function of $\pi_\theta$. This allows the policy optimization to account for the implicit influence of constraint satisfaction, leading to a more informed learning process. The second line in Eq. (9) corresponds to a primal-dual method for solving $\min_{\phi \in \Phi} J(\theta, \phi)$, where $[\cdot]^+$ is a projection operation onto the non-negative real space. Widely used in constrained RL[4], this method can achieve low suboptimality if the world model's parameterization has sufficient representational capacity, as detailed in Paternain et al. (2023). Additionally, the first line in Eq. (9) represents a gradient ascent step corresponding to the objective function $\max_\theta \{ J(\theta, \phi^*(\theta)) \mid \phi^*(\theta) \in \arg\min_{\phi \in \Phi} J(\theta, \phi) \}$, of which the derivation is provided in Appendix F. According to the learning dynamics in Stackelberg games and constrained optimization (Fiez et al., 2020; Paternain et al., 2023), the learning rates in Eq. (9) should satisfy $\eta_\phi^k \gg \eta_\lambda^k \gg \eta_\theta^k$ to ensure convergence.

# 5 PRACTICAL ALGORITHM: ROMBRL

For a discounted finite-horizon MDP with horizon $h$, $J(\theta, \phi) = \mathbb{E}_{\tau \sim P(\cdot; \theta, \phi)} \left[ \sum_{j=0}^{h-1} \gamma^j r_j \right]$, where $\tau = (s_0, a_0, r_0, \cdots, s_h)$ and $P(\tau; \theta, \phi) = d_0(s_0) \prod_{j=0}^{h-1} \pi_\theta(a_j|s_j) P_\phi(r_j, s_{j+1}|s_j, a_j)$. To compute the first-order and second-order derivatives in Eqs. (7) - (9), we have the following theorem:

---

[4]Viewing $P_\phi$ and $\pi_\theta$ as the "policy" and "world model" respectively, $\min_{\phi \in \Phi} J(\theta, \phi)$ is converted into a typical constrained RL problem.

**Theorem 4.** *For an episodic MDP with horizon $h$, let $\Psi(\tau, \theta) = \sum_{i=0}^{h-1} \left( \sum_{j=i}^{h-1} \gamma^j r_j \right) \log \pi_\theta(a_i | s_i)$ and $\Psi(\tau, \phi) = \sum_{i=0}^{h-1} \left( \sum_{j=i}^{h-1} \gamma^j r_j \right) \log P_\phi(r_i, s_{i+1} | s_i, a_i)$. Then, we have:*

$$\nabla_\theta J(\theta, \phi) = \mathbb{E}_{\tau \sim P(\cdot; \theta, \phi)} \left[ \nabla_\theta \Psi(\tau, \theta) \right], \ \nabla_\phi J(\theta, \phi) = \mathbb{E}_{\tau \sim P(\cdot; \theta, \phi)} \left[ \nabla_\phi \Psi(\tau, \phi) \right],$$

$$\nabla_{\phi\theta}^2 J(\theta, \phi) = \mathbb{E}_{\tau \sim P(\cdot; \theta, \phi)} \left[ \nabla_\phi \Psi(\tau, \phi) \nabla_\theta \log P(\tau; \theta, \phi)^T \right], \tag{10}$$

$$\nabla_\phi^2 J(\theta, \phi) = \mathbb{E}_{\tau \sim P(\cdot; \theta, \phi)} \left[ \nabla_\phi \Psi(\tau, \phi) \nabla_\phi \log P(\tau; \theta, \phi)^T + \nabla_\phi^2 \Psi(\tau, \phi) \right].$$

Please refer to Appendix G for the proof of this theorem and the derivatives of $\mathcal{L}(\theta, \phi, \lambda)$.

In a practical algorithm, the expectation terms in Eqs. (10) and (54) are replaced with corresponding unbiased estimators, i.e., the sample means. Estimators of the first-order derivatives can be efficiently computed using automatic differentiation, whose space and time complexity scale linearly with the number of parameters in the models, namely $N_\theta$ and $N_\phi$. However, computing the second-order derivatives, i.e., $\nabla_\phi^2 \Psi(\pi, \phi)$ in $\nabla_\phi^2 J(\theta, \phi)$ and $\nabla_\phi^2 \log P_\phi$ in $\nabla_\phi^2 \mathcal{L}(\theta, \phi, \lambda)$, can be costly. Instead of substituting the second-order terms with $cI$ as in Wang et al. (2021), where $I$ is the identity matrix, we propose the following approximations:

$$\mathbb{E}_{\tau \sim P(\cdot; \theta, \phi)} \left[ \nabla_\phi^2 \Psi(\tau, \phi) \right] \approx -\mathbb{E}_{\tau \sim P(\cdot; \theta, \phi)} \left[ \sum_{i=0}^{h-1} \left( \sum_{j=i}^{h-1} \gamma^j r_j \right) F(s_i, a_i, r_i, s_{i+1}; \phi) \right], \tag{11}$$

$$\mathbb{E}_{(s,a,r,s') \sim P_{\bar{\phi}} \circ \mathcal{D}_\mu(\cdot)} \left[ \nabla_\phi^2 \log P_\phi(r, s' | s, a) \right] \approx -\mathbb{E}_{(s,a,r,s') \sim P_{\bar{\phi}} \circ \mathcal{D}_\mu(\cdot)} \left[ F(s, a, r, s'; \phi) \right],$$

where $F(s, a, r, s'; \phi) = \nabla_\phi \log P_\phi(r, s' | s, a) \nabla_\phi \log P_\phi(r, s' | s, a)^T$ is the Fisher Information Matrix. Thus, we substitute $\nabla_\phi^2 \log P_\phi$ with $-F$, inspired by the fact (Amari, 2016) that $\mathbb{E}_{x \sim P_\phi(\cdot)} \left[ \nabla_\phi^2 \log P_\phi(x) + \nabla_\phi \log P_\phi(x) \nabla_\phi \log P_\phi(x)^T \right] = 0$. In Appendix H, we discuss the approximation errors that arise when using Eq. (11).

The other computational bottleneck is computing the inverse matrix $A^{-1}$ in Eq. (9)[5]. We choose not to use an identity matrix in place of $A^{-1}$ as done in Nikishin et al. (2022). As aforementioned, $A = \nabla_\phi^2 \mathcal{L}(\theta, \phi, \lambda)$ is substituted with its sample-based estimator $\hat{A}$, which is defined as follows:

$$\hat{A} = UV^T - XY^T + ZZ^T, \tag{12}$$

where $U, V \in \mathbb{R}^{N_\phi \times m}$; $X, Y, Z \in \mathbb{R}^{N_\phi \times M}$; $m$ and $M$ represent the number of sampled trajectories and sampled transitions, respectively. Specifically, the $i$-th columns of $U$ and $V$ are given by $\nabla_\phi \Psi(\tau(i), \phi)/\sqrt{m}$ and $\nabla_\phi \log P(\tau(i); \theta, \phi)/\sqrt{m}$, respectively, where $\tau(i)$ denotes the $i$-th trajectory sampled from $P(\cdot; \theta, \phi)$. The $i$-th column of $Z$ is given by $\nabla_\phi \log P_\phi(r_i, s_i' | s_i, a_i) \cdot \sqrt{\lambda/M}$, based on a transition sampled from $P_{\bar{\phi}} \circ \mathcal{D}_\mu(\cdot)$. Additionally, by randomly sampling a trajectory $\tau$ and a time step $t \sim \text{Uniform}(0, h-1)$, we obtain the columns of $X$ and $Y$ as $\left( \sum_{j=t}^{h-1} \gamma^j r_j \right) \nabla_\phi \log P_\phi(s_{t+1}, r_t | s_t, a_t) \cdot \sqrt{h/M}$ and $\nabla_\phi \log P_\phi(s_{t+1}, r_t | s_t, a_t) \cdot \sqrt{h/M}$, respectively. **Given that $m, M \ll N_\theta, N_\phi$, each term in $\hat{A}$ is a low rank matrix and so we can apply Woodbury matrix identity (Fiez et al., 2020) to efficiently compute $\hat{A}^{-1}$.**

To summarize, by leveraging Fisher information matrices and the Woodbury matrix identity, we obtain a close approximation of the gradient update for $\pi_\theta$ (i.e., the first line of Eq. (9)), of which the computational complexity scales linearly with the number of parameters in $\pi_\theta$ and $P_\phi$. In particular, the time complexity with our design and without it are $\mathcal{O}(mN_\theta + M^2 N_\phi)$ and $\mathcal{O}(mN_\theta + MN_\phi^2 + N_\phi^\omega)$, respectively, where $2 \le \omega \le 2.373$. **Please refer to Appendix I for the justification of Eq. (12) and a detailed complexity analysis. The pseudo code and implementation details of our algorithm (ROMBRL) are available in Appendix J.**

## 6 EXPERIMENTAL RESULTS

In this section, we benchmark our algorithm (ROMBRL) against a range of state-of-the-art (SOTA) offline RL baselines across two task sets comprising a total of 15 continuous control tasks. The

---

[5]Since $S$ in Eq. (9) is a scalar, its inverse $S^{-1}$ can be easily computed.

Table 1: Comparison of our algorithm with state-of-the-art offline RL methods on the D4RL MuJoCo benchmark. The abbreviations 'hc', 'hp', and 'wk' denote HalfCheetah, Hopper, and Walker2d, respectively. **To evaluate robustness, we introduce measurement noise into the real MuJoCo dynamics.** Each value represents the normalized score, as proposed in Fu et al. (2020), of the policy trained by the corresponding algorithm. These scores are undiscounted returns normalized to approximately range between 0 and 100, where a score of 0 corresponds to a random policy and a score of 100 corresponds to an expert-level policy. For each algorithm, we report the average score of the final 100 policy learning epochs and its standard deviation across three random seeds. **The best and second-best results for each task are bolded and marked with * respectively.** The last row includes Cohen's $d$ to indicate the significance of our algorithm's improvement over other methods.

| Data Type | Agent Type | ROMBRL (ours) | CQL | EDAC | COMBO | RAMBO | MOBILE |
|---|---|---|---|---|---|---|---|
| random | hc | 39.3* (4.0) | 18.3 (1.2) | 14.9 (7.8) | 5.8 (2.1) | 36.7 (3.4) | **40.3** (2.1) |
| random | hp | **31.3** (0.1) | 10.1 (0.4) | 14.1 (5.6) | 10.3 (3.4) | 25.7* (6.1) | 22.7 (9.1) |
| random | wk | **21.7** (0.1) | 2.3 (1.9) | 0.7 (0.3) | 0.0 (0.1) | 13.1 (3.0) | 17.2* (3.3) |
| medium | hc | 77.5* (1.7) | 48.5 (0.1) | 62.1 (1.0) | 49.6 (0.2) | **78.1** (1.2) | 72.8 (0.5) |
| medium | hp | **102.6** (2.6) | 56.1 (0.4) | 54.8 (38.4) | 71.1 (3.1) | 71.7 (8.2) | 89.9* (16.1) |
| medium | wk | 79.4* (4.2) | 73.9 (0.4) | **83.6** (1.4) | 72.9 (3.1) | 13.7 (5.0) | 71.5 (6.1) |
| med-rep | hc | **73.8** (0.4) | 45.7 (0.3) | 53.2 (0.7) | 46.4 (0.2) | 65.8 (0.4) | 67.6* (4.2) |
| med-rep | hp | **105.3** (0.7) | 94.1 (4.7) | 101.3 (0.6) | 99.5 (1.3) | 97.6 (1.1) | 104.9* (1.8) |
| med-rep | wk | **90.5** (2.4) | 72.2 (17.3) | 83.1 (0.3) | 71.7 (1.9) | 80.9 (1.0) | 85.2* (2.3) |
| med-exp | hc | 88.9* (2.1) | 88.3 (0.5) | 61.2 (6.9) | **90.5** (0.0) | 83.4 (1.4) | 82.9 (5.2) |
| med-exp | hp | **111.9** (0.1) | 109.1 (0.9) | 55.2 (41.3) | 110.1* (0.5) | 83.5 (4.9) | 79.6 (45.5) |
| med-exp | wk | 110.7* (2.6) | 109.9 (0.1) | 60.6 (46.3) | 37.8 (50.9) | 19.5 (20.7) | **113.7** (2.5) |
| Average Score | | **77.7** (0.5) | 60.7 (1.2) | 53.7 (4.6) | 55.5 (3.6) | 55.8 (1.3) | 70.7* (2.4) |
| Cohen's $d$ | | - | 18.9 | 7.4 | 8.5 | 22.9 | 4.1 |

first task set is the widely used D4RL MuJoCo suite (Fu et al., 2020), which includes three types of robotic agents, each with offline datasets of four different quality levels. Since D4RL MuJoCo features deterministic dynamics, we further evaluate on a more challenging set of Tokamak control tasks, where the underlying dynamics are highly stochastic. **To assess the robustness of the policies learned by different algorithms, we introduce noise to the environments' dynamics during deployment. Specifically, at each time step $t$, the state change (in each dimension) $\Delta s_t = s_{t+1} - s_t$ is perturbed with zero-mean Gaussian noise, where the standard deviation is proportional to $\Delta s_t$. In our experiments, we use 5% measurement noise.**

For complete details on our experimental setup, including random seed selection and hyperparameter settings, please refer to Appendix L. For additional results, including computational cost analysis, performance under varying noise levels and training curves, see Appendix M.

## 6.1 RESULTS ON D4RL MUJOCO

Here, we compare ROMBRL with several representative offline RL methods, including two model-free algorithms: CQL (Kumar et al., 2020), EDAC (An et al., 2021), and three model-based algorithms: COMBO (Yu et al., 2021), RAMBO (Rigter et al., 2022), MOBILE (Sun et al., 2023). These baselines are carefully selected: EDAC and MOBILE are top model-free and model-based offline RL algorithms on the D4RL MuJoCo benchmark, according to Sun et al. (2023); CQL and its model-based extension, COMBO, are widely adopted in real-life robotic tasks; RAMBO is explicitly designed to enhance robustness in offline RL.

In Table 1, we report the convergent performance of each algorithm on all tasks. Following the protocol in the D4RL benchmarking paper (Fu et al., 2020), convergent performance is defined as the mean evaluation score over the final 100 policy learning epochs. We present the mean (standard deviation) across runs with different random seeds. **ROMBRL ranks first on 7 out of 12 tasks and second on the remaining 5.** In terms of average performance, ROMBRL significantly outperforms all baselines. To quantify the improvement in average performance, we compute Cohen's $d$ between ROMBRL and each baseline. According to Cohen's rule of thumb (Sawilowsky, 2009), a value of $d \geq 2$ indicates a very large and statistically significant difference. These results demonstrate the superior performance of our algorithm in robust deployment.

Table 2: Comparison of our algorithm against SOTA offline RL methods on standard (noiseless) and noisy D4RL MuJoCo benchmarks. To ensure a fair and reproducible comparison, baseline results are sourced from the official OfflineRL-Kit repository, which reports scores on 9 specific tasks. Consequently, all mean scores presented here are averaged over these 9 tasks, which are composed of 3 environments (HalfCheetah, Hopper, Walker2d) across 3 dataset types (medium, medium-replay, and medium-expert). The 'Performance Drop' metric highlights the superior robustness of our method under deployment noise.

| Metric | ROMBRL (ours) | CQL | EDAC | COMBO | RAMBO | MOBILE |
|---|---|---|---|---|---|---|
| Mean Score (Standard Env.) | 92.8 | 80.4 | 93.0 | 89.3 | 82.7 | **95.9** |
| Mean Score (Noisy Env.) | **93.4** | 77.5 | 68.3 | 72.2 | 66.0 | 85.3 |
| **Performance Drop (%)** ↓ | **-0.6%** | 3.6% | 26.6% | 19.1% | 20.2% | 11.1% |

Table 2 quantitatively demonstrates the core trade-off and ultimate benefit of our robust design. While ROMBRL's performance on the standard (noiseless) benchmark is highly competitive, on par with the SOTA method MOBILE and significantly outperforming the robust baseline RAMBO, it uniquely achieves the top score under noisy deployment conditions where other baselines falter. This slight performance difference in the ideal, noiseless setting is an expected and theoretically justified consequence of our learning objective (Eq. 5), which explicitly optimizes for worst-case robustness across an uncertainty set of models rather than overfitting to a single, known environment. **Ultimately, these results validate that ROMBRL provides a powerful solution to the sim-to-real challenge by drastically improving deployment robustness—as shown by its minimal performance drop—without a significant compromise in standard benchmark performance.**

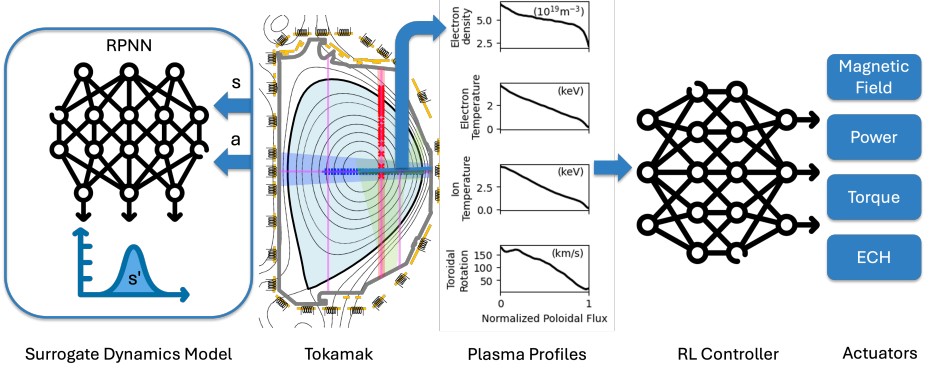

Figure 2: Illustration of the Tokamak Control tasks. The RL controller is trained to apply actuators, such as power and torque, based on the current plasma state, with the goal of driving the plasma toward a target profile. For practical reasons, the real tokamak is replaced with an ensemble of dynamics models trained on operational data from a real device – DIII-D. These models are used to generate data for offline RL and evaluate the trained policies.

## 6.2 RESULTS ON TOKAMAK CONTROL

We further evaluate our algorithm on three target tracking tasks for tokamak control. The tokamak is one of the most promising confinement devices for achieving controllable nuclear fusion, where the primary challengec is confining the plasma, i.e., an ionized gas of hydrogen isotopes, while heating it and increasing its pressure to initiate and sustain fusion reactions (Pironti & Walker, 2005). Tokamak control involves applying a direct actuators (e.g., ECH power, magnetic field) and indirect actuators (e.g., setting targets for the plasma shape and density) to confine the plasma to achieve a desired state or track a given target. This sophisticated physical process is an ideal test bed for our algorithm.

As shown in Figure 2, we use a well-trained data-driven dynamics model provided by Char et al. (2024) as a "ground truth" simulator for the nuclear fusion process during evaluation, and generate a dataset containing 111305 transitions for offline RL. We select reference shots (each of which represents a fusion process) from DIII-D[6], and use trajectories of Ion Rotation, Electron Density, and

---

[6]DIII-D is a tokamak device located in San Diego, California, operated by General Atomics.

Table 3: Comparison of our algorithm with state-of-the-art offline RL methods on the Tokamak Control benchmark. **To evaluate robustness, we inject measurement noise into the tokamak dynamics.** Each value represents the negative episodic tracking error of a specific physical quantity in the reference shot. For each algorithm, we report the average evaluation performance of the final 100 policy learning epochs and its standard deviation across three random seeds. **The best and second-best results for each task are bolded and marked with \* respectively.** The last row reports Cohen's $d$ to quantify the significance of our algorithm's improvement over competing methods. A value of $d \geq 2$ indicates a statistically significant improvement.

| Tracking Target | ROMBRL (ours) | CQL | EDAC | COMBO | RAMBO | MOBILE | BAMCTS |
|---|---|---|---|---|---|---|---|
| $\beta_N$ | -70.9* (0.9) | -78.4 (3.1) | **-63.4** (1.7) | -84.3 (7.6) | -121.1 (19.9) | -133.9 (10.1) | -111.3 (24.3) |
| Density | **-60.0** (1.9) | -87.3 (12.5) | -112.5 (11.1) | -67.0* (3.1) | -81.3 (15.7) | -75.0 (4.3) | -79.6 (13.8) |
| Rotation | **-10.6** (3.7) | -39.2* (10.1) | -95.4 (44.3) | -69.6 (25.9) | -300.3 (260.5) | -257.6 (153.7) | -305.6 (242.6) |
| Average Return | **-47.1** (1.2) | -68.3* (6.8) | -90.4 (11.5) | -73.6 (5.8) | -167.6 (91.6) | -155.5 (47.7) | -165.5 (84.5) |
| Cohen's $d$ | - | 4.3 | 5.3 | 6.3 | 1.9 | 3.2 | 2.0 |

$\beta_N$ within them as targets for three tracking tasks. These are critical quantities in tokamak control, particularly $\beta_N$, which serves as an economic indicator of the efficiency of nuclear fusion. These continuous control tasks are **highly stochastic**, as the underlying dynamics model is an ensemble of recurrent probabilistic neural networks (RPNNs) and each state transition is a sample from this model. **For details about the simulator, and the design of the state/action spaces and reward functions, please refer to Appendix K.**

In addition to the baselines presented in Table 1, we include a recent model-based offline RL method, BAMCTS (Chen et al., 2024a), which has also been evaluated on Tokamak Control tasks. The benchmarking results are shown in Table 3. The evaluation metric is the negative episodic tracking error of the reference shots, computed as the sum of mean squared errors between the achieved and target quantities at each time step. As in previous experiments, we report the average score over the final 100 policy learning epochs as the policy's convergent performance. **The results show that ROMBRL ranks first in 2 out of 3 tasks and second in the remaining one.** Moreover, ROMBRL achieves the best average performance across all three tasks, with notably low variance. Cohen's $d$ further confirms that ROMBRL's improvement over each baseline is statistically significant. In contrast, the performance of several SOTA model-based methods: RAMBO, MOBILE, and BAMCTS, drops significantly in this stochastic and noisy benchmark, showing poor robustness and high variance across runs. The training curves of each algorithm for all tasks are shown as Figure 4 in Appendix K.

## 7 CONCLUSION

In this paper, we propose ROMBRL, a novel offline model-based RL algorithm designed to address two key challenges in the field. First, we aim to jointly optimize the world model and policy under a unified objective, thereby resolving the common objective mismatch issue in model-based RL. Second, we focus on enhancing the robustness of the learned policy in adversarial environments – the key to make offline RL applicable to real-world tasks. To this end, we formulate a constrained maximin optimization problem to maximize the worst-case performance of the policy. Specifically, the policy is optimized to maximize its expected return, while the world model is updated adversarially to minimize it. This optimization is carried out using Stackelberg learning dynamics, in which the policy acts as the leader and the world model as the follower, adapting alongside the policy. We provide both theoretical guarantees and efficient training techniques for our algorithm design. ROMBRL is evaluated on multiple adversarial environments, including 12 noisy D4RL MuJoCo tasks and 3 Tokamak Control tasks, covering both deterministic and stochastic dynamics. ROMBRL outperforms a range of SOTA offline RL baselines with statistical significance, demonstrating strong robustness and potential for real-world deployment.

## REPRODUCIBILITY STATEMENT

To ensure the reproducibility of our research, we have made comprehensive efforts to provide all necessary components. The complete source code for our proposed algorithm, ROMBRL, along with the scripts to replicate all experiments, has been submitted as supplementary material.

A detailed breakdown of our practical algorithm (ROMBRL), including its final pseudo-code (Algorithm 2) and key implementation details, is provided in Appendix J. For experimental validation, our work relies on the publicly available D4RL MuJoCo benchmark . To ensure a fair and consistent comparison, all baseline hyperparameters for the D4RL tasks were adopted from the official OfflineRL-Kit repository . For the non-standard Tokamak Control tasks, a thorough description of the environment, including state-action spaces, reward functions, and dataset generation, is available in Appendix K.

All theoretical claims presented in this paper are supported by rigorous proofs and derivations in the appendix. Specifically, the proof for our main performance guarantee (Theorem 1) is located in Appendix B. The proofs concerning the uncertainty set design for tabular and Gaussian models (Theorems 2 and 3) are provided in Appendices C and D, respectively . Furthermore, the detailed mathematical derivations for the constrained Stackelberg learning dynamics (Equation 9) and the policy gradient formulas (Theorem 4) can be found in Appendix F and Appendix G, respectively. We believe these resources provide a clear pathway for the community to verify and build upon our findings.

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
