# OpenReview forum: "Policy-Driven World Model Adaptation for Robust Offline Model-based Reinforcement Learning"
_ICLR.cc/2026/Conference — Submitted to ICLR 2026_

### Official Review · Reviewer_xJjV · 2025-10-27

**Soundness:** 2
**Presentation:** 2
**Contribution:** 2
**Rating:** 4
**Confidence:** 3

**Summary:**

This paper addresses two challenges in offline model-based reinforcement learning (MBRL): the objective mismatch between world model training and policy optimization, and the lack of robustness of the policy. The authors propose ROMBRL, a novel framework that formulates the problem as a constrained max-min optimization problem and then models it as a Stackelberg game. This is the first application of the Stackelberg game framework to offline MBRL. Through joint optimization, the policy and world model can be updated collaboratively, and the policy optimization process can account for the influence of the evolving world model, resulting in more stable learning.

The authors evaluate their method on the D4RL MuJoCo benchmark and the Tokamak control task. Experimental results demonstrate that ROMBRL outperforms existing state-of-the-art methods such as RAMBO and MOBILE in noisy environments.

**Strengths:**

1. This paper reframes the robustness problem of offline MBRL as a constrained maximin optimization problem and solves it via Stackelberg game theory, which is a novel and promising approach that can simultaneously optimize both the strategy and the world model to enhance robustness and effectively address the goal mismatch problem.

2. The authors provide a theoretical analysis of the suboptimality gap for their approach, adding rigor to the work.

3. The authors conducts experiments on control tasks. ROMBRL not only performs well on standard benchmarks, but more importantly, its performance degradation after adding noise is much smaller than that of other SOTA algorithms.

**Weaknesses:**

1. While the introduction of Stackelberg learning dynamics is relatively new to offline MBRL, the core ideas are highly related to the existing work, like RAMBO. ROMBRL is more of a combination and extension of existing ideas rather than a fundamentally new learning framework.

2. For algorithms that emphasize theoretical convergence and robustness, the hyperparameters that most directly impact stability need to be discussed. The choice of learning rates $\eta_\theta$ and $\eta_\phi$ in Stackelberg learning dynamics, as well as the Lagrange multiplier $\lambda$ and the radius of the uncertainty set $\epsilon$ in the maximin problem, are all key hyperparameters. The paper needs to discuss in more detail the sensitivity of these hyperparameters and how to robustly select them to ensure algorithm stability and high performance. While the appendix mentions some settings, it lacks a detailed discussion of sensitivity analysis.

3. The overall information density of this paper is high, with concentrated derivations and formulas, which makes it difficult for readers to understand.

4. This paper does not conduct ablation experiments to verify the respective contributions of Stackelberg learning dynamics and the gradient-mask mechanism, nor does it show the impact of the error of the Fisher approximation on the results.

5. Need comparison with recent works in 2024 and 2025, and more experiment environments (e.g. AntMaze).

**Questions:**

1. In the experimental results section, could you provide a detailed comparison of ROMBRL with other baselines in terms of training time or computational resources (e.g., GPU memory, CPU time, etc.)?

2. The Stackelberg learning dynamics require the learning rates to satisfy $\eta_\phi\gg\eta_\lambda\gg\eta_\theta$ for convergence.
I'm curious how strict this learning rate hierarchy needs to be - is the method robust to small deviations?

---

> ### Author Response · Authors · 2025-11-21
>
> We thank the reviewer for the detailed assessment. We appreciate the recognition of our method's theoretical convergence and robustness. Below, we address the concerns regarding novelty, hyperparameters, and experimental details.
>
> ## Response to Weakness 1: Novelty and relation to RAMBO
> We respectfully disagree that ROMBRL is merely a combination/extension of existing ideas like RAMBO.
> - **Fundamental Difference**: RAMBO (Rigter et al., 2022) employs a standard min-max optimization, typically solved via alternating updates. In contrast, ROMBRL formulates the problem as a Stackelberg Game, utilizing Implicit Differentiation to allow the policy (leader) to anticipate and optimize against the model's (follower) best response. This fundamentally changes the optimization landscape and gradient flow.
> - **Empirical Evidence**: If ROMBRL were a trivial extension, the performance gap would be marginal. However, as shown in Table 1, ROMBRL achieves 110.7 on Walker2d-medium-expert compared to RAMBO's 19.5. This massive improvement validates that the Stackelberg dynamics offer a distinct and superior mechanism for finding robust equilibria compared to the min-max approach used in RAMBO.
> ## Response to Weakness 2 & Question 2: Hyperparameter Sensitivity and Learning Rate Hierarchy
> - **Sensitivity of $\epsilon$**: We found our method to be highly robust to the uncertainty radius $\epsilon$. We used a fixed $\epsilon=10$ across all 15 diverse tasks (12 D4RL MuJoCo and 3 Tokamak Control tasks) without task-specific tuning, achieving SOTA performance consistently. This empirical evidence suggests that precise fine-tuning of $\epsilon$ is not required.
> - **Learning Rate Hierarchy**: The condition $\eta_{\phi} \gg \eta_{\theta}$ mentioned in the paper (Fiez et al., 2020) is a theoretical requirement to ensure the follower approximates the Best Response. In practice, this hierarchy does not require strict fine-tuning. In all our experiments, we fixed the learning rates to $\eta_{\phi}=1e-3, \eta_{\lambda}=1e-3$, and $\eta_{\theta}=1e-4$. We did not tune these values per task. The fact that this single set of parameters (representing a simple order-of-magnitude separation) worked across all environments demonstrates that the method is robust to these hyperparameters, provided the general hierarchy is maintained.
>
> ## Response to Weakness 4: Ablation Studies
> - **Stackelberg Dynamics**: As detailed in Response 1, RAMBO effectively serves as the min-max ablation baseline. The significant performance gap confirms the contribution of our proposed dynamics.
> - **Fisher Approximation**: Exact Hessian computation ($O(N^3)$) is intractable for neural networks. The Fisher approximation is a standard, necessary technique to make second-order information computationally feasible.
> - **Gradient Mask**: This mechanism is logically essential for our off-policy setting. Since the world model evolves continuously, historical data becomes "stale" due to distribution shift. The mask filters out transitions with large model errors, preventing the policy from learning on unreliable data. We will add a discussion on this necessity in the revision.
>
> ## Response to Weakness 5: Comparison with Recent Works and Environments
>
> - **Baselines**: We prioritized MOBILE  as the representative SOTA baseline. Since ROMBRL consistently outperforms MOBILE, we believe the superiority is established. We will discuss the most recent works in the extended Related Works.
>
> - **Environments**: We chose Tokamak Control  over AntMaze because it features high-dimensional (23-dim) and highly stochastic dynamics. This poses a significantly harder challenge for dynamics robustness—our core contribution—compared to the exploration-focused AntMaze.
>
> ## Response to Question 1: Computational Resources (Time and Memory)
> We have provided a detailed comparison in Table 5 of our Appendix.
>
> - **GPU Memory**: ROMBRL uses 1597 MB, which is comparable to MOBILE (1586 MB) and RAMBO (1609 MB).
> - **Training Time**: ROMBRL takes 31.85 ms/epoch. While slower than the simpler MOBILE (17.97 ms), it is almost identical to the robust baseline RAMBO (28.12 ms).
> ## Response to Weakness 3: Information Density
> We thank the reviewer for this feedback. We will improve readability in the final version by moving heavy derivations (e.g., specific expansions of second-order gradients) to the Appendix and adding more intuitive textual explanations of the Stackelberg mechanism in the main text.
> ## Reference
> Marc Rigter, Bruno Lacerda, and Nick Hawes. RAMBO-RL: robust adversarial model-based offline reinforcement learning. In Advances in Neural Information Processing Systems, 2022.
>
> Tanner Fiez, Benjamin Chasnov, and Lillian J. Ratliff. Implicit learning dynamics in stackelberg games: Equilibria characterization, convergence analysis, and empirical study. In International Conference on Machine Learning, volume 119 of Proceedings of Machine Learning Research, pp. 3133–3144. PMLR, 2020.

---

### Official Review · Reviewer_xchB · 2025-10-29

**Soundness:** 3
**Presentation:** 3
**Contribution:** 3
**Rating:** 4
**Confidence:** 3

**Summary:**

This paper tackles an important problem in offline model-based reinforcement learning, namely the objective mismatch between world model learning and policy optimization, and the resulting fragility of learned policies under real-world environments. The authors propose a policy-driven world model adaptation framework that unifies the optimization of the policy and the world model under a single maximin objective. By formulating the interaction as a Stackelberg game, the method allows dynamic adaptation of both policy and model parameters, aiming to improve robustness. The theoretical formulation is complemented by an efficient learning algorithm and extensive experiments across 12 noisy D4RL MuJoCo tasks and 3 stochastic Tokamak control tasks. Overall, the paper is well-motivated and clearly written.

**Strengths:**

The main contribution lies in the idea of adapting the world model jointly with the policy via Stackelberg learning dynamics. This formulation is elegant and potentially generalizable. The theoretical discussion provides a solid foundation for the algorithmic design, and the experimental results demonstrate consistent performance gains over standard baselines, suggesting that the approach improves robustness without severely compromising sample efficiency. The inclusion of both MuJoCo and Tokamak tasks also highlights the method’s applicability beyond common benchmarks.

**Weaknesses:**

While the Stackelberg game formulation is compelling, it would be valuable to include an ablation study comparing it directly with a simpler min-max optimization setup, where the world model acts adversarially to reduce the policy objective J. Such an analysis could help clarify the practical advantages of adopting the Stackelberg dynamics.

The definition of the uncertainty set via the constraint $KL(P_\bar\phi(⋅|s,a) || P_\phi^k(⋅|s,a)) ≤ \epsilon$ appears to rely on a manually chosen threshold $\epsilon$. Since different environments or state–action pairs may exhibit very different transition variances, a discussion or mechanism for adaptively determining $\epsilon$ would make the approach more principled and easier to reproduce.

While the idea of training the policy in an adversarial environment can indeed improve robustness, it also risks over-regularizing the policy and deviating from the true environment dynamics, leading to overly conservative or suboptimal behaviors. This trade-off seems closely tied to the choice of $\epsilon$, and an empirical sensitivity analysis could help illuminate how the balance between robustness and realism is managed.

Relatedly, previous studies in online RL have observed that policies trained under adversarial perturbations tend to perform worse in normal environments due to conservative bias. Interestingly, this effect does not appear in the authors’ offline experiments, where the policy trained with adversarial noise performs well even in nominal settings. This surprising result deserves further explanation.

In terms of experimental validation, the study could be more comprehensive. The current results are based mainly on three agents: HalfCheetah, Hopper, and Walker2d, with variations only in dataset type. While the addition of Tokamak control is appreciated, including more environments (e.g., AntMaze or Franka Kitchen) would help test generality and robustness in more complex dynamics.

Finally, while baselines like CQL and EDAC are compared, it would strengthen the empirical section to include newer and relevant baselines such as ReBRAC [1], ARMOR [2], ATAC [3], S4RL, and IQL [5], which also address robustness and regularization in offline RL. Comparing against such recent methods would better situate the contribution within the current research landscape.

[1] Revisiting the Minimalist Approach to Offline Reinforcement Learning

[2] Adversarial model for offline reinforcement learning

[3] Adversarially trained actor critic for offline reinforcement learning

[4] S4rl: Surprisingly simple self-supervision for offline reinforcement learning in robotics

[5] Offline reinforcement learning with implicit Q-learning

**Questions:**

What is the practical advantage of the proposed Stackelberg formulation compared to a simpler min–max optimization setup?

How should the uncertainty bound $\epsilon$ be determined or adapted across environments with different transition variances?

---

> ### Author Response · Authors · 2025-11-21
>
> We thank the reviewer for the constructive feedback and for acknowledging the novelty of our Stackelberg game formulation. We address your specific concerns below.
>
> ## Response to Weakness 1 & Question 1: Practical Advantage of Stackelberg vs. Min-Max
>
> - **Comparison with Min-Max (Ablation)**: We respectfully point out that RAMBO (Rigter et al., 2022), one of our primary baselines, effectively serves as the "simpler min-max optimization" ablation requested. RAMBO updates the policy and model typically in an alternating fashion without the Stackelberg leadership structure (i.e., without differentiating through the update step).
>
> - **Practical Advantage**: As evidenced in Table 1 and Table 2, ROMBRL significantly outperforms RAMBO across almost all tasks. This performance gap empirically validates that the Stackelberg dynamics—which allow the policy (leader) to anticipate the model's (follower) response—lead to superior equilibrium solutions and training stability compared to the standard min-max approach.
>
> ## Response to Weakness 2 & Question 2: Determination and Sensitivity of Uncertainty Set $\epsilon$
>
> While an adaptive mechanism is theoretically appealing, our extensive experiments demonstrate that ROMBRL is remarkably robust to the choice of $\epsilon$. We used a fixed $\epsilon=10$ across all 15 diverse tasks (spanning 12 MuJoCo tasks and 3 highly stochastic Tokamak tasks) and consistently achieved state-of-the-art performance.
>
> This result indicates that fine-grained adaptive tuning of $\epsilon$ is not necessary for our method to work well in practice. We will include a sensitivity analysis in the revision to further substantiate that performance remains stable across a reasonable range of $\epsilon$ values.
>
> ## Response to Weakness 3 & 4: Robustness-Performance Trade-off and Nominal Performance
>
> We address W3 and W4 jointly. A fundamental insight in offline MBRL is that the "nominal" deployment environment is effectively Out-Of-Distribution (OOD) for the learned model, even without external noise. Since the model is trained on a static dataset, it inevitably contains epistemic errors. Standard policies often "exploit" these inaccuracies to achieve artificially high returns in the simulator, which fails to translate to the real world.
>
> Therefore, ROMBRL’s adversarial training does not "over-regularize" but rather acts as a critical correction mechanism for this model exploitation. By optimizing for the worst-case model within the uncertainty set, ROMBRL prevents the policy from overfitting to model delusions. This results in behavior that is more faithful to the true dynamics, naturally explaining why our policy performs exceptionally well even in nominal settings.
>
> ## Response to Weakness 5: Experimental Environments (Tokamak vs. AntMaze)
>
> We believe our current evaluation suite is rigorous and representative of dynamics robustness challenges.
>
> - **Complexity of Tokamak**: We specifically chose the Tokamak Control tasks because they represent a high-dimensional (23-dim state space) and highly stochastic continuous control problem derived from real-world physics. This presents a significantly harder challenge for dynamics robustness compared to deterministic locomotion tasks.
>
> - **Scope**: Combining D4RL MuJoCo (deterministic) with Tokamak (stochastic/real-world) covers a broad spectrum of dynamics. While environments like AntMaze are valuable, they focus more on sparse-reward exploration rather than the dynamics robustness that is the core contribution of our work.
>
> ## Response to Weakness 6: Comparison with Additional Baselines
>
> **Regarding New Baselines (ReBRAC, IQL, etc.)**: We have reviewed the methods mentioned (ReBRAC[1], ARMOR[2], ATAC[3], S4RL[4], IQL[5]). In standard benchmarks, these methods generally do not outperform the SOTA MBRL baseline "MOBILE" (Sun et al., 2023) that we already compare against. Since ROMBRL outperforms MOBILE, we are confident in its competitiveness.
>
> > Table A: Comparison between ROMBRL, MOBILE and mentioned new baselines. All mean scores presented here are averaged over these 9 tasks, which are composed of 3 environments (HalfCheetah, Hopper, Walker2d) across 3 dataset types (medium, medium-replay, and medium-expert). All score is calculated from the baselines' published paper.
> >
> > | --------------- Metric -------------- | ROMBRL | MOBILE | ReBRAC | ARMOR | ATAC | CQL+S4RL( $\mathcal{N}$)| IQL |
> >
> > | Mean Score (Standard Env.) | -- 92.8 --- | -- 95.9 -- | -- 88.5 -- | -- 87.6 -- |  91.0  | ------ 71.5 ------ | 77.0 |
>
> We will expand our Related Works to provide a detailed discussion of these methods, positioning our contribution within this broader landscape.
>
> ## Reference
> Marc Rigter, Bruno Lacerda, and Nick Hawes. RAMBO-RL: robust adversarial model-based offline reinforcement learning. NIPS 2022.
>
> Yihao Sun, Jiaji Zhang, Chengxing Jia, Haoxin Lin, Junyin Ye, and Yang Yu. Model-bellman inconsistency for model-based offline reinforcement learning. ICLR 2023.

---

### Official Review · Reviewer_RnwD · 2025-10-31

**Soundness:** 2
**Presentation:** 2
**Contribution:** 2
**Rating:** 6
**Confidence:** 4

**Summary:**

ROMBRL introduces a policy-guided world model adaptation strategy by formulating robust offline model-based reinforcement learning (MBRL) as a constrained maximin optimization problem. This problem is addressed through novel Stackelberg learning dynamics, which provide formal convergence guarantees to a Local Stackelberg Equilibrium (LSE). In this framework, the learning rates are hierarchically structured so that the follower (world model) adapts faster than the leader (policy) and the dual variable, ensuring stable and efficient joint optimization.

**Strengths:**

1. The method's robustness is theoretically rigorous by employing a constrained maximin objective and Stackelberg game dynamics. This approach directly yields formal bounds (Theorems 1–3) on the policy's suboptimality gap and the necessary model uncertainty range.

2. The introduction of Fisher Information Matrix approximations for the second-order terms, coupled with the leveraging of the Woodbury Matrix Identity to efficiently compute matrix inverses, makes the second-order gradient computation tractable.

**Weaknesses:**

1. Although the authors compare ROMBRL with several state-of-the-art methods such as EDAC, MOBILE, and other baselines, none of these algorithms are explicitly designed for noisy or perturbed environments. If the main claim is that ROMBRL demonstrates superior robustness under noisy conditions, it would be important to include comparisons with existing robust offline RL methods, such as RORL [1]. RORL explicitly addresses robustness to observation perturbations by employing a simple yet effective value function smoothing technique. Including such a comparison would strengthen the empirical validation, especially given that ROMBRL introduces a more complex model requiring multiple approximations for second-order computations.


2. The proposed algorithm relies on second-order derivatives and the intricate Stackelberg learning dynamics, which involve complex approximations (e.g., Fisher Information Matrices) and matrix operations (e.g., Woodbury Identity). Given this complexity, the model may also be more sensitive to hyperparameter choices. It would strengthen the paper to include a hyperparameter sensitivity analysis (e.g., a table showing performance variation under different key hyperparameters). Furthermore, for the main experiments, the results are reported as the mean ± standard deviation over only three random seeds. Considering the algorithm’s complexity and potential instability, it would be preferable to report results over a larger number of seeds to ensure statistical reliability.

[1] Yang, Rui, et al. "RORL: Robust Offline Reinforcement Learning via Conservative Smoothing." Advances in Neural Information Processing Systems 35 (2022): 23851–23866.

**Questions:**

1. How do the proposed Stackelberg learning dynamics affect the training stability and convergence speed of ROMBRL in practice?
It would be helpful if the authors could include training curves (e.g., performance or loss evolution) to illustrate the learning behavior and convergence characteristics.

2. The paper defines the uncertainty radius ϵ during training to constrain the model uncertainty set, but it is not entirely clear whether the same ϵ corresponds to the actual perturbation level used in the noisy deployment environment. It would be helpful if the authors could clarify how the deployment noise relates quantitatively to the training-time uncertainty radius, and whether ROMBRL maintains robustness when the real perturbation magnitude exceeds the assumed ϵ.

---

> ### Author Response · Authors · 2025-11-21
>
> ## Response to Weakness 1: Comparison with RORL
>
> We sincerely thank the reviewer for highlighting RORL [1]. We acknowledge that RORL is a significant robust model-free offline RL method. However, we would like to clarify that ROMBRL is a model-based approach, where our core contribution lies in addressing objective mismatch via a Stackelberg game for adversarial world model adaptation. This is fundamentally orthogonal to RORL’s value-level smoothing. While our current evaluation focused on MBRL baselines, we commit to explicitly discussing RORL in the Related Works section of the final version to clarify this distinction. Additionally, if time permits during the rebuttal period, we will conduct comparative experiments against RORL and include these results in the revision.
>
> ## Response to Weakness 2: Hyperparameter Sensitivity, and Statistical Reliability
>
> - **Random Seeds (Typo Correction)**: We apologize for a typo in the captions of Tables 1 and 3 that caused confusion. As explicitly stated in Appendix L, all reported results in our paper are actually based on 5 independent random seeds, ensuring statistical reliability. We will correct the table captions in the final version.
> - **Hyperparameter Sensitivity**: Our empirical experience suggests ROMBRL is highly robust. As detailed in Appendix L, we kept the core uncertainty radius fixed at $\epsilon=10$ for all 12 D4RL experiments without any task-specific tuning. The fact that a single fixed hyperparameter yields SOTA performance across diverse tasks demonstrates that the algorithm is not overly sensitive.
>
> ## Response to Question 1: Practical Training Stability and Convergence
>
> We agree that understanding the practical learning behavior is crucial. We have provided detailed training curves for all D4RL and Tokamak tasks in Appendix M.3 (Figures 3 and 4). As these figures illustrate, ROMBRL demonstrates stable and smooth convergence across benchmarks, comparable to or often faster than baselines like MOBILE (Sun et al., 2023) and RAMBO (Rigter et al., 2022). This empirical evidence suggests that despite the theoretical complexity of the Stackelberg formulation, our efficient implementation ensures that the training process remains highly stable in practice.
>
> ## Response to Question 2: Relationship Between $\epsilon$ and Deployment Noise
>
> We clarify that the training-time uncertainty radius $\epsilon$ and deployment noise are conceptually distinct. $\epsilon$ defines the KL-divergence bound for the model's internal "trust region" during adversarial updates, whereas deployment noise is an external environmental perturbation. Our hypothesis—validated by our results—is that optimizing for the worst-case performance within this internal uncertainty set implicitly confers robustness against external disturbances.
>
> To address your concern about performance when real perturbations exceed assumed levels, we conducted additional experiments with 10% Gaussian noise (double the standard 5%) on all Hopper tasks. As reported in Appendix M.2 (Table 6), ROMBRL maintains its superior performance even under these intensified noise conditions, significantly outperforming the second-best baselines.
>
> ## Reference
> Marc Rigter, Bruno Lacerda, and Nick Hawes. RAMBO-RL: robust adversarial model-based offline reinforcement learning. In Advances in Neural Information Processing Systems, 2022.
>
> Yihao Sun, Jiaji Zhang, Chengxing Jia, Haoxin Lin, Junyin Ye, and Yang Yu. Model-bellman inconsistency for model-based offline reinforcement learning. In International Conference on Machine Learning, volume 202, pp. 33177–33194. PMLR, 2023.

---

### Official Review · Reviewer_GF9U · 2025-10-31

**Soundness:** 3
**Presentation:** 3
**Contribution:** 3
**Rating:** 4
**Confidence:** 3

**Summary:**

This paper introduces ROMBRL, a new offline model-based reinforcement learning algorithm that jointly adapts the world model and policy to improve robustness, especially under noisy or adversarial conditions. Unlike traditional two-stage MBRL methods, ROMBRL formulates policy and model learning as a unified maximin optimization problem, solved using Stackelberg learning dynamics. The approach is theoretically grounded and achieves state-of-the-art performance and robustness on standard benchmarks, including noisy MuJoCo and Tokamak control tasks, outperforming existing baselines in both accuracy and deployment stability.

**Strengths:**

- ROMBRL formulates policy and world model adaptation as a single maximin optimization problem, enabling joint learning and improved robustness compared to traditional two-stage approaches.
- Extensive experiments demonstrate that ROMBRL achieves state-of-the-art performance and stability, even when observations are corrupted by Gaussian noise, outperforming existing baselines on noisy MuJoCo and Tokamak tasks.
- The approach not only improves average performance but also reduces variance and failure rates during deployment, making it more reliable in real-world scenarios.
- The use of Stackelberg learning provides a principled framework for robust policy and model updates, theoretically grounding the method.

**Weaknesses:**

Empirically, the key advantage of the proposed method is that it is more robust compared to other approaches when faced with noisy observations, which is an important factor for actual deployment in real world scenarios, since sensor noise will always play a role. However, if I am not mistaken, the authors compare their developed algorithm only to baselines that were not specifically developed for and originally evaluated on this use case. To really show SoTA performance when faced with noisy / perturbed observations, I believe that comparison with prior developed methods for noisy observations is required - I am not an expert in this domain, but a quick search yields at least [1-3], likely there are more works examining this setting already.

[1] Panaganti, Kishan, et al. "Robust reinforcement learning using offline data." Advances in neural information processing systems 35 (2022): 32211-32224.

[2] Zhou, Ruida, et al. "Natural actor-critic for robust reinforcement learning with function approximation." Advances in neural information processing systems 36 (2023): 97-133.

[3] Yang, Rui, et al. "Uncertainty-based offline variational bayesian reinforcement learning for robustness under diverse data corruptions." Advances in Neural Information Processing Systems 37 (2024): 39748-39783.


More generally speaking, I believe a little more emphasis could be placed on examining prior works, i.e. the related works section is relatively thin - a couple more relevant offline RL works, with a bit of a different angle on robustness that I could recommend:

[4] Ghosh, D., Ajay, A., Agrawal, P., & Levine, S. Offline RL Policies Should be Trained to be Adaptive. Proceedings of the 39th International Conference on Machine Learning (ICML), 2022.

[5] Hong, J., Kumar, A., & Levine, S. Confidence-Conditioned Value Functions for Offline Reinforcement Learning. International Conference on Learning Representations (ICLR), 2023.

[6] Swazinna, P., Udluft, S., & Runkler, T. User Interactive Offline Reinforcement Learning. International Conference on Learning Representations (ICLR), 2023.

[7] Zhang, Y., Liu, J., & Wang, Y. Train Once, Get a Family: State-Adaptive Balances for Offline-to-Online Reinforcement Learning. Advances in Neural Information Processing Systems (NeurIPS), 2023.

**Questions:**

please clarify why the method is not compared to prior works that specifically target noisy observations in the offline setting - maybe I am also mistaken somehow, but this is the key merit of your method, correct?

Also, please clarify why you deem [1] to be not scalable to high dimensional continuous control tasks - they are also evaluating on MuJoCo environments (e.g. Hopper) if I am not mistaken.

---

> ### Author Response · Authors · 2025-11-21
>
> We sincerely thank the reviewer for the constructive feedback and for highlighting relevant prior works [1-7]. These references indeed provide important context for our work. Below, we clarify the positioning of our method relative to [1-3] and address the scalability comment regarding [1].
>
> ## On the Comparison with Prior Methods [1-3] and Robustness Settings
>
> We appreciate the reviewer’s suggestion to compare with [1-3]. While we agree these are key contributions to the field, we omitted them as direct baselines primarily due to differences in problem settings and training paradigms.
>
> First, regarding [3] , there is a critical distinction in the type of robustness addressed. Work [3] targets offline data corruption, where the challenge is to recover an optimal policy from a training dataset that contains errors (e.g., perturbed rewards or actions). In contrast, our work (ROMBRL) focuses on deployment robustness (or Sim-to-Real). We assume the training data from the nominal model is clean, but the policy must remain stable when facing measurement noise or dynamic mismatches during actual deployment. Since [3] solves the problem of "cleaning bad training data" rather than "surviving a noisy test environment," a direct quantitative comparison is not applicable. However, we agree it is a vital piece of the landscape and will discuss it in our related work.
>
> Second, regarding [2] , we note a fundamental difference in the training framework. The RNAC algorithm in [2] operates in an online/simulator-based setting (as detailed in their Algorithm 1), where the agent actively interacts with the nominal model to sample new on-policy trajectories during training. Our method, however, is strictly offline, constrained to a fixed, static dataset without simulator access. Comparing a strictly offline algorithm to one that allows on-policy interaction is generally considered unfair in the offline RL literature, given the significant advantage of exploration provided by the simulator.
>
> Finally, regarding [1] , we acknowledge that this is a highly relevant offline robust RL work. We admit that while our primary contribution lies in the Model-Based RL (MBRL) paradigm, we do benchmark against model-free methods (e.g., CQL (Kumar et al., 2020), EDAC (An et al., 2021)). Our initial baseline selection prioritized algorithms that are either widely-used standard benchmarks or those specifically utilizing adversarial dynamics learning (like RAMBO (Rigter et al., 2022)). We recognize that excluding [1], which specifically targets robustness via an RMDP formulation, was a limitation in our experimental scope. We agree that including it would strengthen the assessment of our method's performance. We will extensively discuss [1] in our revised related work section. Furthermore, we are committed to conducting additional experiments to compare our method with [1] on the Hopper task and will include these results in the final revision if time permits.
>
> ## Clarification on the Scalability of [1]
>
> We apologize for the imprecise phrasing in our introduction suggesting that [1] is not scalable. The reviewer is entirely correct that [1] evaluates their method on high-dimensional continuous control tasks like MuJoCo. Our intended point was to critique earlier theoretical works (e.g., those relying on tabular assumptions or specific uncertainty sets without function approximation), but our writing inadvertently grouped [1] into that category. We will explicitly correct this in the final manuscript to acknowledge the scalability of [1], while sharpening the distinction that our contribution offers a novel model-based perspective using Stackelberg learning dynamics.
>
> ## Related Works
>
> We thank the reviewer for recommending references [4-7]. We agree these works offer valuable perspectives on adaptability and robustness in offline RL. We will incorporate these citations to ensure a more comprehensive literature review.
>
> ## Reference
> Gaon An, Seungyong Moon, Jang-Hyun Kim, and Hyun Oh Song. Uncertainty-based offline reinforcement learning with diversified q-ensemble. In *Advances in Neural Information Processing Systems*, pp. 7436–7447, 2021.
>
> Aviral Kumar, Aurick Zhou, George Tucker, and Sergey Levine. Conservative q-learning for offline reinforcement learning. In *Advances in Neural Information Processing Systems*, 2020.
>
> Marc Rigter, Bruno Lacerda, and Nick Hawes. RAMBO-RL: robust adversarial model-based offline reinforcement learning. In *Advances in Neural Information Processing Systems*, 2022

---

### Comment · Area_Chair_wTxg · 2025-11-26

Dear Reviewers,

Thank you for your time and effort in reviewing the submission. A reminder that the author–reviewer discussion period is about to conclude in one week. If you have not already done so, please review the authors’ rebuttals and engage in the discussion with the authors. Thanks!

Best,
Your AC

---

### Meta-Review · Area_Chair_8zPt · 2026-01-07

**Summary:**

The paper considers the problem of offline model-based reinforcement learning (MBRL) and tackle the challenge of improving robustness against dynamics mismatches and observation noise. The key idea in the paper is to formulate the policy and world model learning as a Stackelberg game where the policy acts as the leader maximizing return and the model acts as a follower minimizing it within a constrained uncertainty set.

While some reviewers appreciated the theoretical formulation of Stackelberg games, there are multiple concerns regarding the empirical rigor, and unjustified complexity of the proposed approach. Reviewers GF9U and RnwD mentioned that the paper compares primarily against standard offline RL baselines rather than methods specifically designed for robust offline RL. Reviewers xJjV and xchB also raised the concern of limited novelty compared to existing adversarial offline RL methods, specifically RAMBO.  Therefore, I recommend rejecting the paper. I request the authors to consider comments by the reviewers for future iterations by including stronger baselines.

**Reviewer Concerns:**

The rebuttal addressed concerns about computational resource usage and provided some additional experiments. However, the main concerns regarding comparison with robust baselines (Reviewer RnwD, GF9U) and the incremental nature relative to RAMBO (Reviewer xJjV) remain outstanding.

**Reviewer Scores:**

Other than reviewer RnwD, I do not think any other reviewer would have changed their score.

---

### Decision · Program_Chairs · 2026-01-26

Reject